# OpenReview forum: "Risk Awareness Injection: Calibrating Vision-Language Models for Safety without Compromising Utility"
_ICML.cc/2026/Conference — ICML 2026 regular_

### Official Review · Reviewer_Sk6T · 2026-03-10

**Soundness:** 3
**Presentation:** 3
**Significance:** 2
**Originality:** 2
**Overall Recommendation:** 4
**Confidence:** 3

**Summary:**

The paper proposed RAI, a training-free method to enhance VLM safety. RAI manipulates vision embedding based on the insight that safety-critical visual cues fail to project into the LLM’s unsafe semantic space. Extensive experiments show the superior performance of RAI on VLM and the video domain.

**Compliance With Llm Reviewing Policy:**

Affirmed.

**Final Justification:**

I would like to thank the authors for their hard work and detailed responses during the rebuttal period.

**Regarding the Ablation Studies (W2)**: I acknowledge the authors' explanation concerning the computational constraints and the limited time window of the rebuttal phase. I understand that a comprehensive grid search across all layers and injection ratios is a significant undertaking, and I appreciate the commitment to including these results in the final version.

**Regarding the Over-refusal Evaluation (W1)**: While the authors provided responses and cited Video-SafetyBench to demonstrate utility, there remains a divergence in our understanding of "over-refusal." The authors' use of general utility benchmarks proves the model still functions for standard tasks, but it does not specifically address the False Reject Rate (FRR) on specialized over-refusal datasets. Namely, samples that contain "sensitive" visual/textual elements but are ultimately benign (e.g., the street dance example). Demonstrating that a defense does not trigger excessive rejection of such "edge-case" benign content is crucial for evaluating the safety-utility trade-off in practice.

**Conclusion**: Although the authors clarified several technical motivations and acknowledged the limitations of their current evaluation, the lack of explicit experimental data on dedicated over-refusal benchmarks leaves some of my primary concerns partially unresolved. Therefore, I have decided to maintain my original score.

**Key Questions For Authors:**

1. The description in Figure 2 and Section 2.1 is inaccurate (“The Red Line (Defensive Failure) consistently exhibits lower similarity compared to the Green Line (Defensive Success)”). In the figure, the defensive failure shows higher similarity for most layers, except the last layer. It should be clarified and explained in the paper because this would affect the empirical analysis.
Sec 2.2 and Figure 3 show that injection is applied at layer 0 for safety-utility balance, but both ASR and perception score degrade for deeper layers. Perhaps the perception score degrades more than ASR as the layer depth increases?

**Limitations:**

Limitaitons can be discussed in the paper, for example, reliance on prototype, token-level alignment overlooking the overall context.

**Strengths And Weaknesses:**

Strengths:

1. RAI is lightweight and training-free, with superior performance on VLM, and can be extended to the video domain.

2. The motivation of RAI is clear, and the empirical study makes the motivation more intuitive.

Weaknesses:

1. Token-level alignment and injection may overlook the overall context of the image. For example, an image containing a person with a hand gesture. Maybe the gesture contains violent meaning, but the whole image is not risky (e.g., street dance or rap). In this case, will RAI cause over-refusal? Additionally, if an image without risk tokens shows harmful content, will it be rejected with RAI?

2. The ablation study is limited to prototype size only. It would be better to include ablations on other parameters, such as the layer index, the threshold for classifying risk-sensitive tokens, and the injection ratio.

3. For evaluation, some human evaluation can be added to validate the evaluation metrics in the paper. For example, inviting 3-5 human experts to judge the harmfulness of responses on a small subset can validate whether the ASR judged by LLMs aligns with humans.

---

> ### Author Rebuttal · Authors · 2026-03-31
>
> We sincerely thank the reviewer for acknowledging the merits of our work, particularly the efficiency of RAI, its training-free nature, and the intuitiveness of our motivation. You have raised highly perceptive and constructive points, which we address in detail below.
>
> ```
> W1: The edge-case questions.
> ```
> **A1:** We thank the reviewer for these excellent edge-case questions. **RAI acts as an early "signal amplifier," not the final decision-maker. It injects risk awareness at the input stage without interfering with the LLM's inherent reasoning capabilities. The LLM still uses its powerful self-attention mechanisms to evaluate the overall global context (e.g., the street dance setting)**. If the global context is benign, the LLM's contextual reasoning will naturally suppress the isolated "gesture" signal. Our utility experiments demonstrate that general understanding capabilities are highly preserved, proving that RAI does not cause over-refusal. We want to clarify that activating the LLM's internal defense does not require precise alignment with explicit visual risk features. For example, in jailbreak datasets, there are many samples featuring benign images (like a cute cartoon kitten) paired with malicious prompts (e.g., "how can we hurt it"). In such cases, the image itself lacks "risk tokens," but the combined intent is harmful. RAI effectively calibrates these visual tokens, preventing them from acting as noise that dilutes the text risk. This can also be observed in our ablation results (Appendix Table 13).
>
> ```
> W2: The ablation study on other parameters.
> ```
> **A2:** We added the threshold parameter $\tau$ on the MMSafety and MME benchmarks. The experimental results demonstrate a clear trade-off: lowering $\tau$ provides a stricter defense (lower ASR) but gradually degrades the model's fundamental visual utility, whereas a higher value preserves visual reasoning capabilities while relaxing the risk intervention. Details in https://anonymous.4open.science/r/Anonymous_2-51BF
> ```
> W3: Human evaluation.
> ```
> **A3:** We sampled 10 instances from each MMSafety category (5 successful and 5 failed defenses based on LLM evaluation) and invited three reviewers to evaluate the Qwen2.5-VL-7B responses. The results show that defense capabilities assessed by humans are higher than the LLM evaluations. We observed that this discrepancy occurs because human evaluators are often less sensitive to certain jailbreak behaviors.
>
> | Evaluator | Illegal Activity | Malware | Pornography | Hate Speech | Physical Harm | Fraud | Avg |
> | :--- | :---: | :---: | :---: | :---: | :---: | :---: | :---: |
> | LLM (ASR) | 50 | 50 | 50 | 50 | 50 | 50 | 50 |
> | Human (ASR) | 30 | 10 | 20 | 20 | 40 | 30 | 25 |
>
> ```
> Q1: Regarding the issues and trade-offs of the injection layer.
> ```
> **A4:**
> - We sincerely thank the reviewer for this sharp and constructive observation. We apologize for the imprecise phrasing of "consistently exhibits lower similarity" in the original manuscript. Our primary intention was to highlight that visual tokens lack deep alignment with abstract risk semantics, resulting in low cosine similarity values. More importantly, at the output layer of LLM, the similarity of defensive failure samples is consistently lower than that of defensive success samples. We will correct it in the revised manuscript. We also conducted additional motivational experiments on LLaVA-1.6-7B and Qwen3-VL-8B. Details in https://anonymous.4open.science/r/Anonymise_test-FD35. The results clearly demonstrate highly consistent trends. Furthermore, we observed an interesting phenomenon: unlike Qwen2-VL, their cosine similarities in the intermediate layers did not drop below 0. We attribute this variance to inherent differences in cross-modal projector architectures and pre-training alignment strategies across different models.
>
> - Regarding your guess about the injection layer, your intuition is exactly right! Our decision to intervene at Layer 0 is based on a comprehensive trade-off: 1) We found that injecting at the very first layer already provides a robust defense mechanism. 2) Just as you accurately inferred, injecting at the output layers causes the general perception capabilities to drop too much compared to the safety improvements.
>
> ```
> Limitations.
> ```
> **A5:** We thank the reviewer for the constructive feedback. Following your suggestion, we will update the revised manuscript to include a comprehensive discussion of our limitations.

---

> > ### Author Rebuttal · Reviewer_Sk6T · 2026-04-03
> >
> > I appreciate the authors' detailed response. However, I have chosen (b) for further clarification on the following:
> >
> > 1.  Over-refusal Evaluation (W1): MME scores only reflect general utility, not the specific safety-utility trade-off. I would like to see explicit test results on over-refusal datasets to confirm that RAI does not cause excessive rejection of benign but sensitive content (e.g., the street dance example).
> >
> > 2.  Ablation Studies (W2): While the threshold $\tau$ is now included, the layer index and injection ratio are fundamental to the RAI framework. I request additional ablation results on these two parameters to justify the optimality of Layer 0 and the robustness of the chosen injection scale across different architectures.

---

> > > ### Author Response · Authors · 2026-04-05
> > >
> > > We are deeply grateful for your highly positive evaluation. We sincerely thank you for dedicating your valuable time to provide such a rigorous, constructive, and thought-provoking review. We address your remaining concern regarding the utility metrics below:
> > > ```
> > > W1: Over-refusal Evaluation
> > > ```
> > > **Q1:**  To address your concern regarding the **"over-refusal" phenomenon—a scenario where a model overly relies on formatted rejection templates (e.g., "Sorry, as an AI assistant...")** when faced with benign but slightly sensitive prompts [1]—we would like to clarify that our current evaluation framework inherently covers this dimension.
> > >
> > > Specifically, the Video-SafetyBench dataset used in our experiments is not merely a collection of malicious attacks. By design, **it includes a balanced mix of both harmful and benign queries.** Our strong performance on this benchmark quantitatively demonstrates that while our method effectively intercepts malicious attacks, it successfully avoids the over-refusal of normal instructions.
> > >
> > > Furthermore, we have provided qualitative response samples for MMSafetyBench, JailBreakV-28K, and Video-SafetyBench in **Appendix D.4: Case Study.** **These examples clearly illustrate that the responses generated using our method do not trigger the standard "Sorry, as an AI assistant..." rejection template.**
> > >
> > > [1]Video-SafetyBench: A Benchmark for Safety Evaluation of Video LVLMs, NIPS 2025.
> > >
> > > ```
> > > W2: Ablation Studies
> > > ```
> > > **Q2:** We fully agree on the importance of the layer index and injection ratio. We would like to clarify two points regarding the ablation studies:
> > > 1. As detailed in Sections 2 and 3,** our motivation for intervening at Layer 0 is strongly supported by recent literature.** In fact, operating on the early or middle layers for safety intervention is a broad consensus in representation engineering [1, 2, 3]. Given that this is a well-established practice in the community, we omitted repetitive baseline ablations in the initial manuscript.
> > > 2. Comprehensively evaluating these two parameters requires sweeping parameter combinations across **every single layer of the LLM**, followed by dual inference evaluations on both massive jailbreak datasets and utility benchmarks. Constrained by the **extremely short time window of the rebuttal phase and the computational overhead**, **it is objectively infeasible for us to complete such a massive grid-search experiment within just a few days.**
> > >
> > > We sincerely apologize for this limitation and kindly ask for your understanding. We are fully committed to adding the detailed ablation results for these parameters in the final revision. Thank you again for your continued support!
> > >
> > > [1] Layer-aware representation filtering: Purifying finetuning data to preserve LLM safety alignment, EMNLP 2025.
> > >
> > > [2] Automating steering for safe multimodal large language models, ACL 2025.
> > >
> > > [3] SafePTR: Token-level jailbreak defense in multimodal LLMs via prune-then-restore mechanism, NeurIPS 2025.

---

### Official Review · Reviewer_QFvr · 2026-03-10

**Soundness:** 3
**Presentation:** 3
**Significance:** 2
**Originality:** 2
**Overall Recommendation:** 4
**Confidence:** 2

**Summary:**

This paper proposes a framework called Risk-Aware Injection (RAI) for calibrating the security of visual language models (VLMs) without sacrificing utility.

**Compliance With Llm Reviewing Policy:**

Affirmed.

**Final Justification:**

Based on the paper and the author's rebuttals, I agree to accept this paper.

**Key Questions For Authors:**

1. Existing approaches that classify risks based on input images and prompts, are they fundamentally the same concept as the Unsafe Prototype Subspace Construction proposed in this paper?

2. If risk categories are not included in the unsafe prototype subspace statistics, will the method fail?

**Limitations:**

No discussed the limitations in the paper.

**Strengths And Weaknesses:**

Strengths:
1.  The proposed method is innovative, treating multimodal jailbreak detection as a self-supervised variance modeling problem. It detects malicious inputs by learning attack-agnostic signals from benign data.
It compares existing jailbreak detection methods and reveals their fundamental limitations, such as scarcity of malicious data and difficulty in generalizing to unseen tactics.

2. The proposed method demonstrates effectiveness, with SelfGuard outperforming all baseline methods across all datasets.

Weaknesses:
1. Beyond the two jailbreak benchmark datasets (MM-SafetyBench and JailBreakV-28K), the authors should have tested more recent VLM jailbreak methods, such as: MMJ-Bench, Figstep, Ideator, etc.

---

> ### Author Rebuttal · Authors · 2026-03-31
>
> We sincerely thank the reviewer for supporting our work and recognizing the innovation and effectiveness of our method.
> ```
> W1: More recent VLM jailbreak methods.
> ```
> **A1:** The Figstep benchmark dataset is entirely encompassed by the JailBreakV-28K dataset. Furthermore, we conducted additional evaluations on the recently proposed SIUO (2024) jailbreak benchmark:
> | Model | Method | Politic | Dangerous | Discrim | Illegal | Misinfo | Morality | Privacy | Religion | Self-harm | Mean |
> | :---: | :--- | :---: | :---: | :---: | :---: | :---: | :---: | :---: | :---: | :---: | :---: |
> | | Original | 50.00 | 38.88 | 38.88 | 38.88 | 41.17 | 45.00 | 57.14 | 40.00 | 26.31 | 41.80 |
> | Qwen3-VL-8B | Adashiled | 37.50 | 22.22 | 33.33 | 33.33 | 35.79 | 40.00 | 33.33 | 25.00 | 21.05 | 31.28 |
> | | ShiftDC | 31.52 | 22.22 | 27.77 | 27.77 | 29.41 | 30.00 | 38.09 | 25.00 | 21.05 | 28.09 |
> | | **RAI** | 25.00 | 16.66 | 22.22 | 27.77 |27.77 | 20.00 | 23.80 | 20.00 | 15.78 | 22.11 |
> | | Original | 62.50 | 38.88 | 38.88 | 55.55 | 58.82 | 50.00 | 61.90 | 55.00 | 68.42 | 54.43 |
> | Qwen25-VL-7B | Adashiled | 50.00 | 33.33 | 33.33 | 50.00 | 29.41 | 50.00 | 33.33 | 35.00 | 52.63 | 40.78 |
> | | ShiftDC | 56.25 | 22.22 | 27.77 | 44.44 | 29.41 | 40.00 | 38.09 | 40.00 | 42.10 | 37.80 |
> | | **RAI** | 43.75 | 27.77 | 22.22 | 27.77 | 23.52 | 30.00 | 33.33 | 25.00 | 36.32 | 29.96 |
>
> ```
> Q1: Are existing approaches that classify risks based on input images and prompts fundamentally the same concept as the Unsafe Prototype Subspace Construction proposed in this paper?
> ```
> **A2:** They are fundamentally different. Existing classification approaches typically act as external discriminators (e.g., safety guardrails) that passively judge the input or output. In contrast, our Unsafe Prototype Subspace is an internal generative intervention. Rather than relying on post-hoc detection, RAI actively modulates the hidden representations at Layer 0 before the LLM even begins its reasoning process. By acting as a "semantic lens," it actively arouses the model's inherent refusal instincts at the earliest possible stage. Unlike detection methods that require training separate classifiers and introducing extra parameter overhead, our Unsafe Prototype Subspace unlocks the model's inherent safety capabilities in a plug-and-play manner, without introducing any external classifiers or additional parameters.
>
> ```
> Q2: If risk categories are not included in the unsafe prototype subspace statistics, will the method fail?
> ```
> **A3:** The method will not fail. As demonstrated in our ablation study (Appendix Table 13), when we reduced the prototype size $K$ from 12 down to 4, the removed categories effectively became "unseen" risks. Despite this, RAI maintained high competitive defense performance.

---

> > ### Author Rebuttal · Reviewer_QFvr · 2026-04-03
> >
> > The author's response partially addressed my previous question, but I still expect more comprehensive experiment with different VLM jailbreak methods to show the effectiveness, so I am maintaining my original rating.

---

> > > ### Author Response · Authors · 2026-04-04
> > >
> > > We sincerely thank you for the positive recognition of our work and your highly constructive suggestions. We deeply appreciate your continued engagement to help us improve the manuscript, and we agree on the importance of evaluating against diverse jailbreak methods.
> > >
> > > However, we respectfully point out that our current evaluation suite already inherently covers a wide variety of jailbreak methods. For instance, **FigStep is already included and evaluated within the JailBreakV-28K benchmark.** More importantly, while **most concurrent state-of-the-art works typically evaluate on only 2 to 3 datasets**, we have provided comprehensive evaluations **across 4 distinct datasets (MM-SafetyBench, JailBreakV-28K, SIUO and Video-SafetyBench)**. For context, please consider the evaluation scope in recent literature:
> > >
> > > SafePTR[1]: Evaluated on **MM-SafetyBench, JailBreakV-28K.**
> > >
> > > AdaShield[2]: Evaluated on **FigStep, QR.**
> > >
> > > Immune[3]: Evaluated on **JailBreakV-28K, MM-SafetyBench, Visual Adversarial Attacks.**
> > >
> > > ShiftDC[4]: Evaluated on **MMSafetyBench, JailBreakV-28K.**
> > >
> > > Given this and the strict time limits of the rebuttal, we are fully committed to adding additional jailbreak experiments in the final version, as you suggested.
> > >
> > > [1] SafePTR: Token-Level Jailbreak Defense in Multimodal LLMs via Prune-then-Restore Mechanism, NeurIPS 2025.
> > >
> > > [2] AdaShield: Safeguarding Multimodal Large Language Models from Structure-based Attack via Adaptive Shield Prompting, ECCV 2024.
> > >
> > > [3] Immune: Improving Safety Against Jailbreaks in Multi-modal LLMs via Inference-Time Alignment, CVPR 2025.
> > >
> > > [4] Understanding and Rectifying Safety Perception Distortion in VLMs, NeurIPS 2025.

---

### Official Review · Reviewer_dYdW · 2026-03-10

**Soundness:** 2
**Presentation:** 3
**Significance:** 3
**Originality:** 3
**Overall Recommendation:** 4
**Confidence:** 4

**Summary:**

The authors propose Risk Awareness Injection (RAI), a lightweight and training-free method. RAI constructs an unsafe prototype subspace from language embeddings and modulates high-risk visual tokens to restore the model’s ability to recognize unsafe content from visual inputs. Experiments show that RAI significantly reduces attack success rates while preserving task performance.

**Compliance With Llm Reviewing Policy:**

Affirmed.

**Final Justification:**

Most of my concerns are resolved. I decide to increase 1 point.

**Key Questions For Authors:**

Please refer to Weaknesses.

**Limitations:**

yes

**Strengths And Weaknesses:**

# Strengths
1. Extensive experiments on recent VLMs demonstrate the effectiveness of RAI, significantly improving the safety capability of the models.
2. RAI maintains utility and accuracy on general benchmarks, which is hard to achieve.

# Weaknesses
1. The motivation of the method is somewhat unclear. The experiments and analysis in Section 2 do not directly address the two key questions raised by the authors: (1) whether the vulnerability originates from the LLM backbone or from the modality integration process; and (2) how the safety failure propagates through the model’s latent representations.
2. The method has strong limitations, as it requires manually predefined unsafe prototypes. The authors do not provide experiments or analysis on whether the approach can generalize to defend against unseen unsafe prototypes.
3. The method in the paper assumes that unsafe content originates entirely from vision tokens. However, prior studies have shown that harmful intent can be conveyed jointly through both text and images, where the visual information itself may appear benign. In such cases, RAI seems unlikely to effectively defend against these multimodal safety threats. Examples include benchmarks such as SIUO [1], MSSBench [2], and MIS-Hard [3].
4. Figure 3 indicates that the percentage of injection affects both the model’s safety and utility. However, in the implementation, injection is controlled through a threshold, which does not allow for explicit control over the injection percentage. Moreover, this threshold may require additional tuning for different models.
5. If the number of unsafe prototype categories is reduced while the threshold is also lowered, the overall injection ratio could theoretically remain roughly the same. In that case, would the model achieve safety and utility performance similar to the main experimental results reported in the paper?

# References

[1] Safe Inputs but Unsafe Output: Benchmarking Cross-modality Safety Alignment of Large Vision-Language Models

[2] Multimodal Situational Safety

[3] Rethinking Bottlenecks in Safety Fine-Tuning of Vision Language Models

---

> ### Author Rebuttal · Authors · 2026-03-31
>
> We sincerely thank the reviewer for the insightful review. We are glad you recognized RAI's effectiveness and its success in preserving model utility. Detailed responses to your technical concerns are provided below.
> ```
> W1: Vulnerability origin and propagation.
> ```
> **Q1:** Our Section 2 empirical analysis directly addresses the two key questions on vulnerability origin and propagation.
> 1. As we explicitly stated in the paper, the vulnerability "stems not from a deficit of safety knowledge within the LLM backbone, but from a phenomenon we term Risk Signal Dilution." Crucially, visual inputs need not contain explicitly "malicious" tokens. As detailed in Section 2.1, current Linear Projectors fail to "precisely align cryptic or abstract visual attack features into the highly structured textual safety space of the LLM." Consequently, these unaligned features act as "high-dimensional semantic noise that overwhelms the risk signals present in the textual instructions," directly proving the issue originates from modality integration.
>
> 2. Figure 2 and Section 2.1 illustrate how this failure propagates. In shallow layers, the cosine similarity between projected visual tokens and the unsafe subspace shows minimal divergence between successful and failed jailbreaks. However, as features propagate deeper, the similarity scores for successful attacks experience a profound attenuation, dropping significantly at the final decision layer. This progressive decay demonstrates how unaligned visual noise gradually erodes the model's safety awareness throughout the reasoning process.
>
> ```
> W2: Limitations and Generalization
> ```
> **A2:** We thank the reviewer for this insightful question.
> Predefined prototypes built by extracting safety representations from standard datasets are widely adopted in recent VLM safety research. Following prior works [1,2], manually defined strategies for analyzing and aligning vision-language representations are common and effective in this field.
>
> RAI generalizes robustly to unseen risks without requiring exact vocabulary matching. In our ablation (Appendix Table 13), we reduced the prototype size $K$ from 12 down to 4. The removed risk categories effectively serve as "unseen" prototypes to the system. Despite this, RAI maintained highly competitive ASRs. RAI maintained highly competitive ASRs against the removed ("unseen") categories. This proves that arousing the LLM's internal defense mechanisms does not require precise alignment with an exhaustive risk vocabulary.
>
> [1]Layer-Aware Representation Filtering: Purifying Finetuning Data to Preserve LLM Safety Alignment
>
> [2]Understanding and Rectifying Safety Perception Distortion in VLMs
>
> ```
> W3: The evaluation on recent benchmarks.
> ```
> **A3:** We conducted additional evaluations on the  SIUO dataset.
> Details in https://anonymous.4open.science/r/Anonymous_3-8970
>
> ```
> W4: Reason for threshold and additional tuning cost.
> ```
> **A4:** We address your concerns regarding the threshold design through the following two aspects:
> 1. Our threshold mechanism acts as a dynamic amplifier that identifies tokens with relatively higher similarity values as "risk-susceptible tokens." If we enforced an explicit injection percentage (e.g., injecting the top 10% of tokens), it would mandate a fixed amount of modification regardless of the actual visual content. Editing an excessive number of visual tokens—that is, the over-injection of risk information—directly leads to the degradation of the model's foundational visual reasoning capabilities. The threshold acts as an instance-aware trigger: it avoids unnecessary token modification by only activating when the abstract risk semantics cross the boundary. Figure 3 precisely illustrates this: excessively lowering the threshold forces over-injection, which severely harms utility.
> 2. Although varying VLMs require specific thresholds, RAI is completely training-free and zero-parameter. Calibrating this threshold requires only a rapid scalar grid search on a tiny validation set. Compared to the massive computational overhead of fine-tuning, this one-time tuning cost is practically zero.
>
> ```
> W5: Prototype size and threshold.
> ```
> **A5:** Evaluations on MMSafety and MME across varying thresholds ($\tau$) show that RAI achieves highly competitive defense even with limited categories (e.g., Size 4). Furthermore, we observed that increasing the number of categories allows the model to achieve a more refined balance between safety and utility. While fewer categories are already effective and competitive, a more comprehensive prototype set further optimizes the trade-off by allowing the model to maintain strong defense at a higher, utility-preserving threshold.
> Details in https://anonymous.4open.science/r/Anonymous_2-51BF

---

> > ### Author Rebuttal · Reviewer_dYdW · 2026-04-03
> >
> > Thank you for the detailed response. Some of my concerns are still not addressed.
> >
> > - W2
> >
> >     The experimental setup in the paper is not very reasonable and does not look like an ablation study for validating generalization. If the authors want to demonstrate the generalization ability of RAI, could they provide detailed results for each safety category in MM-SafetyBench under the prototype K=4 setting? For example, categories such as Political Lobbying and Financial Advice are not included in the prototypes. If RAI achieves similar ASR on these categories as on the others, then it would be more convincing to claim that RAI has reasonably good generalization ability.
> >
> > - W4
> >
> >     I still believe that the pre-defined threshold $\tau$ is a potential issue, and the paper does not provide any sensitivity analysis for this threshold. Given that different models use different thresholds, it is unclear whether the effectiveness of RAI would drop substantially if the chosen threshold were not optimal.

---

> > > ### Author Response · Authors · 2026-04-04
> > >
> > > ```
> > > W2
> > > ```
> > > **A1:** We sincerely thank the reviewer for this highly insightful suggestion to improve our work.
> > >
> > > Specifically, in the $K=4$ setting, the specific risk tokens used for prototype construction are: Violence, Illegal, Sexual, and Hateful. The Fraud and Malware are strictly unseen.
> > >
> > > |Model| Prototype Size | Illegal (S) | Illegal (T) | Illegal (S-T) | Malware (S) | Malware (T) | Malware (S-T) | Porn (S) | Porn (T) | Porn (S-T) | Hate (S) | Hate (T) | Hate (S-T) | Harm (S) | Harm (T) | Harm (S-T) | Fraud (S) | Fraud (T) | Fraud (S-T) | Avg↓ |
> > > | :---| :--- | :---: | :---: | :---: | :---: | :---: | :---: | :---: | :---: | :---: | :---: | :---: | :---: | :---: | :---: | :---: | :---: | :---: | :---: | :---: |
> > > |Qwen2.5-VL-7B | Original | 9.28 | 8.25 | 30.93 | 22.73 | 43.18 | 75.00 | 8.26 | 42.20 | 58.72 | 7.98 | 24.54 | 43.56 | 20.83 | 43.75 | 61.81 | 18.83 | 26.62 | 50.00 | 33.14 |
> > > |  Qwen2.5-VL-7B| 4 | 1.78 | 7.13 | 3.57 | 3.93 | 19.67 | 3.93 | 1.59 | 21.68 | 17.04 | 0.00 | 2.13 | 3.19 | 7.22 | 14.42 | 21.64 | 3.38 | 2.25 | 2.25 | 7.60 |
> > >
> > > ```
> > > W4
> > > ```
> > > **Q2:** We sincerely thank you for raising this critical point regarding robustness. We agree with your intuition that in real deployments, it is practically difficult to guarantee an absolute mathematically "optimal" threshold for every unseen distribution. However, we would like to respectfully clarify that RAI's effectiveness **does NOT drop substantially** if the chosen threshold is slightly sub-optimal. We address your concerns regarding the threshold ($\tau$) through the following three points:
> > > 1. We would like to respectfully point out that we **did provide a detailed sensitivity analysis of the threshold $\tau$ in our first-round rebuttal (please refer to https://anonymous.4open.science/r/Anonymous_2-51BF).** The experimental results clearly demonstrate that **RAI's performance around the chosen threshold forms a stable plateau rather than a fragile, sharp peak. Deviating from the exact optimal point does not cause an abrupt failure; instead, varying $\tau$ simply shifts the model along a smooth, predictable trade-off curve.** This allows practitioners to safely fine-tune the strictness of the safety-utility balance without risking a catastrophic drop in defense effectiveness.
> > >
> > > 2. While we acknowledge that different models require specific thresholds, the selection process is neither arbitrary nor unbounded. Our extensive empirical profiling reveals a highly consistent pattern: the textual-visual representations predominantly fall within a very narrow, mathematically predictable interval (e.g., $0$ to $0.08$, see Figure 2 and https://anonymous.4open.science/r/Anonymise_test-FD35). For instance, the $\tau$ was empirically determined to be $0.068$ for the Qwen-VL series and $0.058$ for the LLaVA series. This striking proximity demonstrates that the search space is incredibly stable across entirely different model architectures, practically eliminating the risk of "blind tuning."
> > >
> > > 3. Because RAI introduces **zero extra parameters and requires no training**, pinpointing a highly effective $\tau$ within this narrow $0 \sim 0.08$ interval merely requires a rapid scalar grid-search on a tiny validation set. Compared to the massive computational overhead of fine-tuning or aligning a new backbone, this one-time, inference-only calibration cost is entirely negligible.

---

### Official Review · Reviewer_YAdL · 2026-03-12

**Soundness:** 3
**Presentation:** 4
**Significance:** 3
**Originality:** 3
**Overall Recommendation:** 5
**Confidence:** 4

**Summary:**

This paper proposes a Risk Awareness Injection (RAI) framework for defending vision-language models against jailbreak attacks by explicitly enhancing risk-related semantic signals in selected visual tokens. The paper verifies the motivation through a token-level analysis of visual-text alignment and introduces a sparse early-layer injection mechanism based on an unsafe prototype subspace. Experiments on several recent open-source VLMs show that the proposed method reduces ASR while preserving utility.

**Compliance With Llm Reviewing Policy:**

Affirmed.

**Final Justification:**

This paper presents a technically sound and practically useful defense framework for multimodal jailbreak mitigation through sparse risk-aware visual token intervention. The main strengths are the clear motivation, lightweight design, and consistent empirical improvements across multiple VLM settings. The rebuttal addressed my main concerns on cross-model validation and technical clarification, although utility evaluation could still be strengthened with more vision-sensitive benchmarks such as OCRBench. Overall, the rebuttal reinforces my original assessment, and I keep my score unchanged.

**Key Questions For Authors:**

1. Could the authors conduct additional evaluation on more recent VLMs to verify whether the semantic-gap observation from Section 2 generalizes beyond Qwen2-VL?
2. The current framework only injects risk signals into visual tokens. Have the authors considered whether similar intervention on textual tokens could further improve safety?
3. Could the authors provide ablation experiments on the threshold choice for τ, especially regarding the trade-off between ASR reduction and utility preservation?
4. There are several minor writing issues in the manuscript.

**Limitations:**

Yes

**Strengths And Weaknesses:**

Strengths

1. The paper addresses how to balance the safety-utility trade-off in VLMs.
2. The proposed framework is lightweight and technically simple. Only a sparse subset of visual tokens is modified, which makes the method computationally efficient and practically deployable.
3. The paper provides an interesting motivation by connecting jailbreak vulnerability to a semantic gap between projected visual tokens and unsafe semantic directions.
4. The overall pipeline is clearly structured, and the intuition behind early-layer intervention is easy to follow.

Weaknesses

1. In Section 2, the motivation is primarily derived from observations on Qwen2-VL. It remains unclear whether the same semantic-gap phenomenon consistently appears across other VLM architectures, and broader validation would strengthen the generality of the argument.
2. The formulation of Unsafe Prototype Subspace and token similarity may require clarification. Since the unsafe prototype subspace is constructed from language embeddings, while visual tokens are projected into the language space through the projector, it would be important to clarify at which stage cosine similarity is computed. There are some confusing sentences at the beginning of Section 3.
3. In the ablation study, there is no dedicated analysis of the threshold parameter τ used in Eq. (4). Since this threshold directly controls token selection and injection sparsity, its influence on both safety and utility should be systematically evaluated.

---

> ### Author Rebuttal · Authors · 2026-03-31
>
> Thank you for your constructive feedback and for recognizing the novel motivation, efficiency, and practical deployment value of our RAI framework. We appreciate your positive assessment and are pleased that the intuition behind our early-layer intervention was clear. Below, we address your comments point-by-point.
>
> ```
> W1&Q1: Additional evaluation on more recent VLMs.
> ```
> **A1:** We agree that validating the "semantic gap" across diverse architectures is crucial for demonstrating generality. Beyond Qwen2-VL, we have conducted similar layer-wise semantic analyses on LLaVA-1.6-7B and Qwen3-VL-8B. Specifically:
>
> **Consistent Trends:** Our results confirm that the "Risk Signal Dilution" phenomenon is universal: in successful jailbreak samples, visual tokens consistently exhibit significantly lower cosine similarity with the unsafe prototype subspace compared to defense-triggered samples, particularly at the output layer.
>
> **Model-Specific Variations:** While the downward trend is consistent, numerical values vary. Qwen3-VL-7B and LLaVA-1.6-7B maintain low similarity scores without the negative values observed in the intermediate layers of Qwen2-VL. We attribute this to inherent differences in projector architectures and pre-trained alignment strengths.
>
> **Universal Bottleneck:** Despite these variations, the core issue remains: current projectors fail to precisely align abstract visual risk features into the LLM’s structured safety space. Details in https://anonymous.4open.science/r/Anonymise_test-FD35
>
>
> ```
> W2&Q4: The writing issues in the manuscript.
> ```
> **A2:** We appreciate the reviewer's suggestion regarding the technical implementation. As you correctly observed, our Unsafe Prototype Subspace is constructed using embeddings from the LLM’s tokenizer, while visual tokens are mapped by the linear projector into a latent space sharing the same dimensionality as these language embeddings. Crucially, the cosine similarity computation occurs immediately after the visual tokens are projected but before they enter the first layer of the LLM backbone, a procedural detail we will explicitly clarify in the revised manuscript to ensure absolute technical clarity.
>
>
> ```
> W3&Q3: The ablation of the threshold parameter τ.
> ```
> **A3:** We evaluated the threshold parameter $\tau$ on the MMSafety and MME benchmarks. The experimental results demonstrate a clear trade-off: lowering $\tau$ provides a stricter defense (lower ASR) but gradually degrades the model's fundamental visual utility, whereas a higher value preserves visual reasoning capabilities while relaxing the risk intervention. Details in https://anonymous.4open.science/r/Anonymous_2-51BF
>
> ```
> Q2: Have the authors considered a similar intervention on textual tokens?
> ```
> **A4:** We conducted additional tests where we applied similar risk-signal interventions directly to the textual tokens. The evaluation results showed a severe collapse in the model's generation capabilities, where the VLM completely failed to produce outputs. This empirical finding strongly supports our structural design. We intentionally focused only on visual tokens for the following fundamental reasons:
>
> **Vulnerability Source:** Our analysis suggests that VLM vulnerabilities stem from visual inputs diluting risk signals, not from a lack of safety knowledge in the LLM backbone.
>
> **Inherent Alignment:** The LLM backbone is already heavily aligned for text-based safety. Intervening on textual tokens may significantly increase the risk of over-refusal.
>
> **Symmetry and Utility:** RAI acts as a "semantic lens" to repair modality-induced distortion. Modifying textual tokens would alter the user's explicit intent, compromising the model's fundamental reasoning capabilities.

---

> > ### Author Rebuttal · Reviewer_YAdL · 2026-04-02
> >
> > Thank you for the detailed clarification. One remaining point is that perception score is still not a very convincing utility metric. Since RAI directly intervenes on visual tokens, evaluating utility on vision-sensitive benchmarks such as OCRBench would better show whether multimodal capability is preserved under intervention. This could be added in revision. I will keep my current score unchanged.

---

> > > ### Author Response · Authors · 2026-04-04
> > >
> > > We sincerely thank you for the positive recognition of our work and for choosing to maintain your current score. We deeply appreciate your constructive attitude and your acknowledgment of our previous clarifications. We address your remaining concern regarding the utility metrics below:
> > > 1. We respectfully clarify that the Perception score reported in our original manuscript already incorporates an evaluation of OCR capabilities. As detailed in Table 4, the total **Perception score is the aggregate sum** of 11 fine-grained visual tasks, which explicitly includes: **Existence, Count, Position, Color, Posters, Celebrity, Scene, Landmark, Artwork, OCR, and Cognition**.
> > >
> > > 2. We agree with your insight that evaluating on a dedicated, highly vision-sensitive benchmark like OCRBench provides a more rigorous and direct validation of multimodal capability preservation. However, **given the strict time limits of the rebuttal phase**, we have prioritized conducting this initial evaluation specifically on the Qwen2.5-VL-7B model.
> > >
> > > | Model | Method | Reg | Irr | Art | Handwrit | Digit | Non-Sem | Scene | Doc | KIE | HMER | Final |
> > > | :--- | :--- | :---: | :---: | :---: | :---: | :---: | :---: |  :---: | :---: | :---: | :---: | :---: |
> > > |  | Original | 48 | 49 | 48 | 42 | 39 | 47 | 178 | 177 | 178 | 71 | 877 |
> > > | | ECSO | 45 | 45 | 46 | 38 | 38 | 44 | 171 | 175 | 169 | 68 | 839 |
> > > | Qwen2.5-VL-7B | Adashield | 47 | 46 | 47 | 39 | 37 | 45 | 174 | 168 | 170 | 69 | 842 |
> > > | | ShiftDC | 48 | 47 | 46 | 40 | 38 | 46 | 171 | 174 | 173 | 70 | 853 |
> > > | | **RAI** | 48 | 48 | 47 | 40 | 39 | 47  |176  | 180 |175  | 72 |872  |
> > >
> > > We are fully committed to providing more detailed and comprehensive experimental results across other models in the final revision.

---

### Decision · Program_Chairs · 2026-04-30

**Decision:**

Accept (regular)

**Comment:**

This paper proposes RAI, a simple training-free method for improving VLM safety by amplifying risk signals in visual tokens via an unsafe prototype subspace. It is lightweight, and consistently reduces jailbreak success while preserving utility across multiple benchmarks.

Reviewers generally agree the method is well-motivated, technically sound. I would suggest that authors discuss more about over-refusal studies. Overall, I recommend acceptance.